# Factors Relating to a Safety Culture in the University Perinatal Center: The Nurses’ and Midwives’ Perspective

**DOI:** 10.3390/ijerph19169845

**Published:** 2022-08-10

**Authors:** Janina Ribelienė, Jūratė Macijauskienė, Rasa Tamelienė, Aušrelė Kudrevičienė, Irena Nedzelskienė, Aurelija Blaževičienė

**Affiliations:** 1Department of Nursing, Lithuanian University of Health Sciences, 44307 Kaunas, Lithuania; 2Faculty of Nursing, Lithuanian University of Health Sciences, 44307 Kaunas, Lithuania; 3Clinical Department of Neonatology, Lithuanian University of Health Sciences, 44307 Kaunas, Lithuania; 4Department of Dental and Oral Diseases, Lithuanian University of Health Sciences, 44307 Kaunas, Lithuania

**Keywords:** patient safety culture, nurse, midwife, Safety Attitudes Questionnaire

## Abstract

**Background**: According to The Joint Commission, a culture of safety is a key component for achieving sustainable and safe health care services, and hospitals must measure and monitor this achievement. Promoting a patient safety culture in health services optimally includes midwifery and nursing. The first aim of this study is to assess the University Perinatal Center’s staff members’ perceptions of safety culture. A second aim is to identify how the perceptions of safety culture actors are related to the socio-demographic characteristic of the respondents. **Methods:** A descriptive, cross-sectional, correlational design was applied in this study. Registered nurses and midwives were recruited from the University Perinatal Center in Lithuania (N = 233). Safety culture was measured by the Safety Attitudes Questionnaire (SAQ). **Results:** The mean scores of the responses on the 6 factors of the SAQ ranged from 3.18 (0.46) (teamwork climate) to 3.79 (0.55) (job satisfaction) points. The percentage of positive responses to the SAQ (4 or 5 points on the Likert scale) ranged from 43.2% to 69.0%. The lowest percentage of the respondents provided positive responses to the questions on perception of management and teamwork climate, while the highest percentage of the respondents provided positive responses to the questions on job satisfaction. Perception of management positively correlated with safety climate (r = 0.45, *p* < 0.01) and working conditions (r = 0.307, *p* < 0.01). Safety climate positively correlated with job satisfaction (r = 0.397, *p* < 0.01) and working conditions (r = 0.307, *p* < 0.01). Job satisfaction positively correlated with working conditions (r = 0.439, *p* < 0.01). **Conclusion:** Evaluating the opinions of the safety climate among nurses and midwives who work at the University Perinatal Center showed that teamwork climate and perception of management are weak factors. Therefore, stakeholders should organize more training about patient safety and factors that affect patient safety.

## 1. Background

Establishing a culture of safety in health care has been a global health policy priority. The Joint Commission includes establishing a safety culture as a critical component for achieving highly reliable and safe care and requires hospitals to measure and monitor safety culture in an ongoing fashion [1]. 

Patient safety culture has been defined by various researchers. Summarizing the definitions of all researchers, it can be argued that a patient safety culture includes shared values, beliefs, and norms in health care institutions and how they influence the behaviours of the health care workers [2]. Patient safety culture is a part of organizational culture. It relates specifically to the values and beliefs concerning patient safety within health care organizations [3].

Safety culture is determined by the symbiosis of three organizational factors: (1) environmental structures and processes within the organization, (2) the attitudes and perceptions of workers, and (3) the safety-related behaviours of individuals [4].

Developing a systematic approach to patient safety culture should allow health care workers to work in a learning environment that provides quality care for patients and promotes quality improvement in the system. Workers also need to feel safe to speak up and report unsafe practices, errors, or adverse events [5,6].

Patient safety is greatly influenced by the peculiarities of teamwork and staff communication, as well as by management. Many researchers have pointed out that most departments are understaffed and have heavy workloads, which often causes stress, emotional exhaustion, and burnout among nursing staff [7,8]. 

Nurses’ perceptions of patient safety culture have been seen to increase their compliance with safe nursing practices, thus creating a safe medical environment [9,10,11]. Nurses often deal with critical situations and should demonstrate critical thinking and independent decisions, which are influenced by knowledge, attitude, leadership, and communication [12]. Promoting a patient safety culture in health services requires an optimal role for nursing.

Assessing the culture of patient safety and implementing impact measures has become necessary in the obstetrics area [13]. However, there is very little evidence about this area. 

A study by Brazilian researchers showed that maternity hospital professionals highlight poor communication between shifts; errors are rarely used to promote a safety culture among professionals. Additionally, adverse events are not reported, and staff believe it is not helpful for the authority to assess patient safety culture [10]. Australian researchers also revealed that patient safety culture in maternity care units is only moderate and required improvement across all six safety culture domains [11].

Therefore, assessing the patient safety culture in obstetric care is of fundamental importance because maternal and child health (MCH) is a priority on the global development agenda, such as the Sustainable Development Goals (SDGs) promoted by the United Nations [14].

The first aim of this study is to assess University Perinatal Center staff perceptions of the safety culture. A second aim is to identify how the perceptions of safety culture actors are related to the socio-demographic characteristic of the respondents.

## 2. Methods

### 2.1. Setting and Sample

A descriptive, cross-sectional, correlational design was applied in this study. At the University Perinatal Center, there are a total of 240 working midwives and nurses, of whom 233 participated in our study. The University Perinatal Center has obstetrics and gynaecology clinics (labour and delivery, obstetrics, gynaecology, and adult intensive care units, and a 140-bed operating room) and neonatal clinics (a neonatal intensive care unit (NICU) and a 92-bed intermediate care unit). This population of nurses and midwives served as the same sample pool for another study [15]. 

### 2.2. Instruments

The final questionnaire package consisted of several sections. First, midwives were asked various demographic questions (sex, age). Second, respondents were asked about work-related characteristics (position, workload, the most common shift, years of work according to specialty, years of work in the unit and hospital, and working hours during the week). The questionnaires used were the Safety Attitudes Questionnaire (SAQ), the Organizational Culture Assessment Instrument (OCAI), the Hospital Survey on Patient Safety Culture (HSOPSC), and the Nursing Role Perception Questionnaire (NRPQ). Safety Attitudes Questionnaire (SAQ) factors and definitions are proved in Table 1. 

This paper will report on the outcomes measured by the Safety Attitudes Questionnaire (SAQ). The Safety Attitudes Questionnaire (SAQ) is a widely used tool for assessing patient safety, has good psychometric properties, and shows associations with clinical outcomes [16,17,18,19]. The Safety Attitudes Questionnaire (the short form) was designed for the staff of obstetrics units, labour and delivery units, intensive care units, and operating rooms. The questionnaire was translated into Lithuanian and was tested for validity and reliability by the State Health Care Accreditation Agency under the Ministry of Health. The questionnaire consists of 36 items assessing 6 safety culture factors: teamwork climate, safety climate, job satisfaction, stress recognition, perceptions of unit and hospital management, and working conditions. Eight Lithuanian hospitals participated in the pilot study. The results of the study revealed that the instrument is suitable for use in assessing patient safety culture in hospitals across the country [20].

The respondents were asked to rate the answers on a 5-point Likert scale, ranging from 1 (strongly disagree) to 5 (strongly agree). A positive score consisted of the mean percentage of the respondents who scored 4/5 points on the items within a factor. A negative score consisted of the mean percentage of the respondents who scored 1/2 points on the items within a factor. All negatively worded items (2, 11, and 36) were reversed.

The evaluation of the internal consistency for the six factors of the SAQ yielded Cronbach’s α of 0.71; the working conditions factor had the lowest Cronbach’s α.

### 2.3. Data Collection

The study data were collected from 1 May to 1 July 2017. One of the authors (J.R.) personally distributed the questionnaires to all eligible midwives and nurses at the University Perinatal Center. A total of 233 questionnaires were distributed. All questionnaires were correctly completed and suitable for further analysis (response rate, 100%). The confidentiality of the subjects was ensured, as they were not asked to give their names, surnames, or addresses. The survey was conducted following the principles of voluntariness, anonymity, and confidentiality. This study was approved by the Kaunas Regional Biomedical Research Ethics Committee passed at the committee session on 2 February 2017 (protocol No. BE-2–15). The respondents gave written consent to participate in the study after familiarising themselves with its aim and methods.

### 2.4. Statistical Analysis

Statistical analysis was performed using IBM SPPS Statistics 22 software [21]. The data are presented as absolute value, percentages, means (M), and standard deviations (SD). The frequency of item values, means, and standard deviations of item correlations were calculated to determine item characteristics. For the assessment of the internal consistency and reliability of the SAQ, Cronbach’s alpha was calculated. The normality (Kolmogorov–Smirnov) test was applied to all data. The groups were compared using the Mann–Whitney (U) test and Fisher’s criterion. The chi-square test was used to test statistical hypotheses about the independence of the features. Spearman’s correlation coefficient was calculated to evaluate the relationship between the parameters. The level of statistical significance of 0.05 was set for testing the hypotheses. 

## 3. Results

In this manuscript, we provide results about safety attitudes from nurses, midwives, anaesthetists, and intensive care and operating room nurses. In our previous work, we published results about hospital- and unit-level patient safety from the two units’ health care workers’ perspectives [15].

In total, 26.6% of the respondents worked in the midwife position, 30.5% were anaesthetists and intensive care nurses; 7.7% were operating room nurses, and 34.8% were general practice nurses (Table 2).

### 3.1. Safety Culture from Nurse and Midwives’ Perspective

The mean score of the responses on the six factors of the SAQ ranged from 3.18 (0.46) to 3.79 (0.55) points. The highest mean score was observed for the job satisfaction factor, while the lowest was observed for the teamwork climate factor. The mean response score for the perception of management factor was significantly higher among the respondents working in obstetrics and gynaecology units than in those working in neonatology units (*p* = 0.002) (Table 3).

Spearman’s correlation analysis showed that the factors of teamwork climate and safety climate had moderate or strong correlations with job satisfaction and perception of management.

Working conditions was statistically significantly correlated with four factors (not teamwork climate). The perception of management factor had moderate or strong correlations with teamwork climate, safety climate, and job satisfaction (see Table 4).

### 3.2. Demographic and Professional Factors Related to Safety Culture

Our research data revealed statistically significant positive attitudes about the SAQ domain of teamwork climate. Safety climate and perception of management were more associated with nurses who worked in anaesthesia and intensive care units and operating rooms compared with midwives and nurses. The shift on which respondents worked was also related to SAQ scores. Midwives had statistically significantly more positive attitudes towards stress recognition compared with anaesthesia and intensive care unit and operating room nurses (Table 5).

## 4. Discussion

Over the last decade, there has been an increasing focus on assessing and improving the culture of safety in health care [22,23]. According to The Joint Commission, a culture of safety is a key component for achieving sustainable and safe health care services, and hospitals must measure and monitor this achievement [1]. 

Obstetrics, gynaecology, and neonatal intensive care are complex and multidimensional activities in which the measurement of safety culture is a priority in health care institutions. Timely emergency care in obstetric services, adequate communication, good management, teamwork, and a safe environment for patients can reduce adverse events by 15% [24].

Culture surveys such as the SAQ are becoming an important component of managing patient safety. These surveys can help hospital management understand the impact of various interventions across the entire hospital system as well as individual units within the hospital system [25,26,27].

The findings of the present study reveal that the lowest evaluation received was for teamwork climate domain (3.18). An explanation for this discrepancy with other research data [11,27] could be that midwives and nurses in East Europe were long considered physician assistants and it has taken time for physicians to adjust to the roles of midwives and nurses changed takes time for physicians.

The highest score in our study was for job satisfaction, with scores of 3.9 and 3.8 for the midwives and nurses, respectively. Brazilian and Turkish researchers analysed the attitudes of midwives, nurses, and doctors toward a safety culture and revealed that their respondents rated job satisfaction the highest [11,26].

The domain of work conditions allows for the evaluation of the strengths of a hospital’s organizational structure, such as training and supervision for new personnel and the process of providing necessary information about diagnostic and therapeutic decisions [28,29]. The results of our study, and those conducted in Chinese and Danish hospitals [18,30], showed that 57% of respondents agreed or strongly agreed with these statements. Increasing workload causes fatigue, which means that the quality of work deteriorates and the risk of errors increases, especially in complex and critical situations. In our study, 50.8% of the respondents agreed with the statements about the factor of stress recognition. Similar results were obtained by Kristensen and co-authors [19]. However, in studies conducted in China and the USA, the percentages of positive responses to this factor were only 20.5% and 30%, respectively. We found that the mean score of responses in this factor was 3.28 (0.65). According to the results of other studies, the mean response score for this factor ranged from 2.79 to 3.62 [13,28,29].

Our study showed that in the studied health care institutions providing services in obstetrics, gynaecology, and neonatology, the weak factors were teamwork climate and perception of management (less than 50% of positive responses according to the Likert scale). More than half of the respondents believed that managers underestimated their efforts, there was not enough staff to ensure proper care for all patients, they were not timely informed about any changes that might affect their work, and the departments did not pay enough attention to patient safety. Similar results were obtained by researchers analysing this factor in hospitals in Denmark (42.6%) [18] and the UK (37%) [31]. In contrast, according to a survey conducted by Zhao and co-authors in China, only 28.3% of the respondents disagreed with the perception of management factor [12].

The assessment of safety culture was highly dependent on sociodemographic and occupational factors. Greek scholars who analysed midwives’ attitudes towards safety culture with the SAQ instrument we used found that experienced midwives rated the following domains higher than did than midwives with less experience: teamwork, safety climate, job satisfaction, and working conditions [28,31].

A study by researchers in England involving staff who had worked in the maternity unit sought to identify factors that affect the Safety Attitudes Questionnaire (SAQ) scores. The study results showed that the six Safety Attitudes Questionnaire domains were related to years of working experience, gender, and professional position. Women and respondents with more extended work experience had more negative attitudes towards safety culture. Meanwhile, men rated all five areas of safety culture better [32,33]. The results of our study revealed that attitudes towards safety culture were determined by respondent’s position and shift. Meanwhile, length of service did not influence the assessment of safety culture.

This study had several limitations. The subjects were nurses and midwives working at only one perinatal centre in Lithuania and providing obstetrics, gynaecology, and neonatology services. For this reason, the obtained results do not reflect the safety climate of all hospitals that provide obstetrics, gynaecology, and neonatology services in Lithuania. The survey data were also collected from spring to summer months when the staff was on vacation, and thus the results were affected by higher workloads and overtime.

### Implications for Practice and Key Message

Culture surveys such as the SAQ are becoming an important component of managing patient safety. These surveys can help hospital management understand the impacts of various interventions across the entire hospital system as well as individual units within the hospital system.

This study provides some first evidence about midwives and nurses’ attitudes and understanding of patient safety culture; our study is one of the first studies that evaluate the attitudes of perinatal centre midwives and nurses toward safety culture. It fills the gap in knowledge about what factors from the point of view of midwives and nurses are important in creating a safe culture in a hospital and allows us to raise certain hypotheses and inspire further research.

In general, the safety climate in the overall study group in the perinatal centre was reported as fairly good (mean score 3.0 and above). Moreover, based on the data, we can create a strategy for improvement or maintenance.

## 5. Conclusions

The evaluation of the attitudes towards safety climate among nurses and midwives working in a health care institution that provides services in obstetrics, gynaecology, and neonatology showed that the decisive factors were job satisfaction, safety climate, work conditions, and stress recognition. The majority of the nurses and midwives at the perinatal centre liked their job, believed that patient safety was ensured in their units, and thought that new staff were trained well and that all the necessary diagnostic and treatment information was available. The weak factors were teamwork climate and perception of management. The respondents felt that managers did not value their efforts enough, that the units were understaffed, and that the workload was high. Focusing on these factors might help to improve patient safety and to reduce the number of adverse events.

## Figures and Tables

**Table 1 ijerph-19-09845-t001:** SAQ factor definitions and example items [16].

Scale: Definition	Example Items
**Teamwork climate:** perceived quality of collaboration between personnel.	I have the support I need from other personnel to care for patients. The physicians and nurses here work together as a well-coordinated team.
**Job satisfaction:** positivity about the work experience.	This office is a good place to work. Working in this office is like being part of a large family.
**Perceptions of management**: approval of managerial action.	The management of this office supports my daily efforts. Nurse input is well received in this office.
**Safety climate**: perceptions of a strong and proactive organisational commitment to safety.	During emergencies, I can predict what other personnel are going to do next. I am encouraged by my colleagues to report any patient safety concerns I may have.
**Working conditions**: perceived quality of the work environment and logistical support (staffing, equipment, etc.).	This office does a good job of training new personnel. All the necessary information for diagnostic and therapeutic decisions is routinely available to me.
**Stress recognition**: acknowledgement of how performance is influenced by stressors.	When my workload becomes excessive, my performance is impaired. I am less effective at work when fatigued.

**Table 2 ijerph-19-09845-t002:** Respondents’ characteristics.

	**% (n)**
**Total number of respondents**	**100 (233)**
Obstetrics and Gynaecology clinics	56 (131)
Neonatology clinics	44 (102)
**Sex**	**100 (233)**
Male	0.9 (2)
Female	99.1 (231)
**Staff position**	**100 (233)**
Midwife	26.6 (62)
Anaesthetist and intensive care nurse	30.5 (71)
Operating room nurse	7.7 (18)
Nurse	34.8 (81)
**Professional experience**	**100 (233)**
Less than 1 year	1.3 (3)
1–5 years	12.0 (28)
6–10 years	5.6 (13)
11–15 years	4.3 (10)
16–20 years	12.0 (28)
21 years or more	64.8 (151)
**Experience in the unit**	**100 (233)**
Less than 1 year	3.0 (7)
1–5 years	16.3 (38)
6–10 years	7.7 (18)
11–15 years	9.4 (22)
16–20 years	14.2 (33)
21 years and more	49.4 (115)

**Table 3 ijerph-19-09845-t003:** SAQ factor, mean scores, and standard deviations of the obstetrics and gynaecology clinics and neonatology clinics.

Factors	All Respondents N = 233	Respondents of Obstetrics and Gynaecology Clinics N = 131	Respondents of Neonatology Clinics N = 102	*p*
	Mean (SD)	Mean (SD)	Mean (SD)	
Teamwork Climate	3.18 (0.46)	3.19 (0.45)	3.18 (0.46)	0.63
Safety Climate	3.5 (0.43)	3.51 (0.42)	3.49 (0.44)	0.79
Job Satisfaction	3.79 (0.55)	3.88 (0.50)	3.77 (0.61)	0.34
Stress Recognition	3.28 (0.65)	3.25 (0.63)	3.24 (0.68)	0.66
Perception of Management	3.22 (0.69)	3.25 (0.63) *	3.06 (0.72)	0.002
Working Conditions	3.5 (0.5)	3.5 (0.48)	3.50 (0.54)	0.70

PRR—Positive response rate. SD—standard deviation. *—statistically significant.

**Table 4 ijerph-19-09845-t004:** Inter-correlations of the six factors.

Factors	Teamwork Climate	Safety Climate	Job Satisfaction	Stress Recognition	Perception of Management	Working Conditions
Teamwork Climate						
Safety Climate	0.45 *					
Job Satisfaction	0.206 ***	0.397 ***				
Stress Recognition	0.328 ***	0.095	0.080			
Perception of Management	0.199 ***	0.447 ***	0.310 ***	0.094		
Working Conditions	0.054	0.370 ***	0.439 ***	0.171 ***	0.307 ***	

* Correlations are significant at *p* < 0.01.

**Table 5 ijerph-19-09845-t005:** SAQ factor evaluation by sociodemographic and professional characteristics.

Domains	Staff Position			*p* Value
	Midwife	Anaesthetists and Intensive Care Nurse/Operating Room Nurse	Nurse	
	M (SD)	M (SD)	M (SD)	
Teamwork climate	3.0 (0.4) *	3.3 (0.4) *	3.2 (0.5)	F = 7.168, df = 2, *p* = 0.001; * *p* < 0.05
Safety climate	3.5 (0.4)	3.6 (0.4) *	3.4 (0.5) *	F = 3.596, df = 2, *p* = 0.029; * *p* < 0.05
Job satisfaction	3.9 (0.4)	3.8 (0.6)	3.8 (0.5)	F = 0.963, df = 2, *p* = 0.383
Stress recognition	3.3 (0.5)	3.2 (0.8)	3.4 (0.5)	F = 1.852, df = 2, *p* = 0.159
Perception of management	3.2 (0.7) **	3.6 (0.5) **	2.9 (0.6) *, **	F = 26.613, df = 2, *p* < 0.001 **, * *p* < 0.05
Working conditions	3.4 (0.5)	3.6 (0.5)	3.5 (0.5)	F = 1.335, df = 2, *p* = 0.265
	**Professional experience**			
	**<10**	**11–20**	**>20**	** *p* ** **-+Value**
	**M (SD)**	**M (SD)**	**M (SD)**	
Teamwork climate	3.2 (0.4)	3.0 (0.4)	3.2 (0.5)	F = 1.123, df = 2, *p* > 0.05
Safety climate	3.4 (0.4)	3.3 (0.3)	3.5 (0.5)	F = 2.556, df = 2, *p* > 0.05
Job satisfaction	3.7 (0.5)	3.6 (0.3)	3.8 (0.6)	F = 1.335, df = 2, *p* > 0.05
Stress recognition	3.3 (0.6)	3.0 (0.7)	3.3 (0.6)	F = 1.844, df = 2, *p* > 0.05
Perception of management	3.1 (0.7)	3.0 (0.6)	3.2 (0.7)	F = 2.513, df = 2, *p* > 0.05
Working conditions	3.4 (0.5)	3.5 (0.4)	3.5 (0.5)	F = 1.156 df = 2, *p* > 0.05
	**Shift**			
	Morning	Night/Afternoon	Mixed (morning, afternoon and night shift)	*p* value
	M (SD)	M (SD)	M (SD)	
Teamwork climate	3.1 (0.5)	3.3 (0.3)	3.1 (0.4)	F = 2.335, df = 2, *p* > 0.05
Safety climate	3.5 (0.4)	3.4 (0.3)	3.4 (0.4)	F = 2.598, df = 2, *p* > 0.05
Job satisfaction	3.8 (0.6)	3.7 (0.5)	3.7 (0.5)	F = 1.068, df = 2, *p* > 0.05
Stress recognition	3.4 (0.7) *	3.3 (0.7)	3.1 (0.6)	F = 4.673, df = 2, *p* < 0.05
Perception of management	3.2 (0.7)	3.2 (0.7)	3.1 (0.7)	F = 0.607, df = 2, *p* > 0.05
Working conditions	3.5 (0.5)	3.5 (0.6)	3.4 (0.5)	F = 0.618, df = 2, *p* > 0.05

* *p* < 0.05; ** *p* < 0.001 - level of statistical significance.

## Data Availability

The data presented in this study are available on request from the corresponding author. The data are not publicly available due to the privacy of the organization.

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
