# Peer review of "Factors Relating to a Safety Culture in the University Perinatal Center: The Nurses’ and Midwives’ Perspective"

_ijerph, 2022, doi:10.3390/ijerph19169845_

Round 1
Reviewer 1 Report
Generally, this is an interesting piece of work that examines patient safety culture and the attitudes of nurses and midwives with regards to the factors that comprise safety culture. The data collection was executed in an ethical manner and the sample size is sufficient for the analyses presented. Furthermore, the authors are aware of the limitations of their study which is always a positive sign. Despite the practical and policy significance of this work, I would like to raise some points that the authors should consider.
1. The literature review is quite sparse. The definition of safety culture seems incomplete since it should also include aspects such as work relations or job environment. Additionally, it should be delineated from safety climate.
2. The authors define safety culture but do not describe its components. Suddenly, in page 3 the authors present 6 factors based on SAQ. How does each of these factors contribute to safety culture? Maybe table 1 should come earlier in the introduction as it includes the definitions of each of the factors.
3. The connection between safety culture with organisational culture should be outlined since much of the literature on safety culture is based on organisational culture.
4. Make sure to clearly state what your research questions are. I’m not sure that the statement in lines 78-79 accurately describes what you’re looking at in the results.
5. By authors’ admission, some reliability coefficients were extremely low (below the .7 cut-off). This means that the items comprising the respective factors are not cohesive and, thus, the respective factors should not be considered in the analyses. Report reliability coefficient per factor in Table 1.
6. The normality test was applied; however, its results are not reported. I surmise that, given the use of non-parametric correlations, the authors found significant violations of the normality assumption. If so, then why have been ANOVAs (F ratios) implemented in some cases?
7. In lines 137-138, the chi-square and the Mann-Whitney tests are mentioned; however, I don’t see the results of these tests being reported.
8. Were there any missing data and how were they handled?
9. The authors compare two separate groups, namely midwives and nurses or gynaecology and neonatology. Have you checked whether the socio-demographics (Table 2) of these two groups differ significantly? If so, then I would be sceptical of any differences given that these may be driven by the demographic variations.
10. In table 5, the groups are also split between midwives, anaesthetists and intensive care nurse/Operating room nurses, and nurses. Is there any reason why the authors split the nurse group in two separate categories? Doesn’t make much sense since there isn’t some reported literature in the introduction indicating such differences.
11. Overall, a lot of comparative analyses were reported without some substantial theoretical background for these specific analyses. Can the authors include some model and/or theory and/or previous empirical work backing these types of analyses?
12. In the discussion, the authors discuss their findings in terms of previous studies’ findings. Maybe move some of that content to the introduction as an appetiser of what’s to come in the results.
13. Factor in any changes due to COVID-19. The study was conducted before COVID. How could COVID change these views?
14. All in all, make sure to clarify what is it that this research adds to extant knowledge? What is the gap/ or discussion/ or debate that this work adds to?
Author Response
We appreciate your invitation to respond to the editors and reviewers of our manuscript and we are pleased to take a chance to improve it. We made changes and improvements according to the comments and summarized the editor’s and the reviewers’ comments and our responses to each point in the attached table.
In this manuscript, we provided results about Safety Attitudes from nurses, midwives, anaesthetists, intensive care nurses’ and operating room nurses' perspectives.
We believe that the paper has now been much improved, and the key messages clarified in response to the constructive suggestions of the editor. Thank you for considering this revised version for publication.
Yours sincerely,
Aurelija Blaževičienė, Professor
Head of Nursing Department, Lithuanian University of Health Sciences,
address: Eivenių 4, Kaunas, LT-44307
cell: +370 682 45938
work: +370 37 327147
Email address: aurelija.blazeviciene@lsmuni.lt

Reviewer 2 Report
I think the manuscript deals with exciting and appropriate issues, i.e., patient safety culture, nurse, midwife, and safety attitudes questionnaire. It looks pretty organised, but a few of my observations follow the given structure below:
First, the abstract looks too long and overloaded with information. It may annoy the reader to read further to explore the entire paper. Please try to make it more reader-friendly and concise.
Second, this paper needs more detailed treatment of the recent (2021-22) empirical literature on patient safety culture, nurse, midwife, and safety attitudes questionnaire. I can see the further prospect or development of the applied theories in the literature to justify their current standing and relation to this study. For example, I cannot see any reference of papers from 2020-21 to understand the current standing of the studied variables (e.g., presenteeism, productivity, nurses, influenza and hospital occupancy). Hence, the context and literature of the empirical analysis should be updated with the following significant papers. For example, you may see Edgar et al., 2021; Elmwafie et al., 2022; Haque, 2021; and Schram et al., 2022. for your further literature.
Third, the research gap is not clear enough. It is essential to understand how this study - what is examined here - offers new insights into the literature? For example, how patient safety culture, nurse, midwife, and safety attitudes questionnaire be managed effectively for higher healthcare performance?
Fourth, the sampling and the justification of (233) need the detail of the participants (e.g., who, why and how? context of Lithuania?).
Fifth, when the paper considers presenteeism with safety culture and healthcare professionals, there is an angle of the role of HRM or HR departments. Hence, a bit of discussion focusing on HRM overcoming presenteeism and considering the workload and higher productivity should be followed. Please see Haque (2018) in this regard.
Finally, I found the discussion of practical implications is a bit unclear. How should practices be designed or aligned within the healthcare industry in Lithuania? What new insight can this paper offer for the importance of patient safety culture and nurse-midwife wellbeing? How should patient safety culture be managed? It can be better to see clear and new takeaways here. I see the prospect, but this paper can be better with its solid practical contributions.
Good luck!
References:
Edgar, D., Middleton, R., Kalchbauer, S., Wilson, V., & Hinder, C. (2021). Safety attitudes build safety culture: Nurse/midwife leaders improving health care using quantitative data. Journal of Nursing Management, 29(8), 2433-2443.
Haque, A. (2018). Strategic human resource management and presenteeism: a conceptual framework to predict human resource outcomes. New Zealand Journal of Human Resources Management, 18(2), 3-18.
Elmwafie, S. M., Abdallah, A. I., & Abduallah, R. M. (2022). Impact of Safety Guidelines on Nurses’ Knowledge regarding Incidents and Nurses’ Safety Attitude at Neonatal Intensive Care Unit. Tanta Scientific Nursing Journal, 25(22), 152-163.Schram, A., Paltved, C., Lindhard, M. S., Kjaergaard-Andersen, G., Jensen, H. I., & Kristensen, S. (2022). Patient safety culture improvements depend on basic healthcare education: a longitudinal simulation-based intervention study at two Danish hospitals. BMJ open quality, 11(1), e001658.
Haque, A. (2021). The effect of presenteeism among Bangladeshi employees. International Journal of Productivity and Performance Management. DOI: 10.1108/IJPPM-06-2020-0305
Author Response

(The authors gave the same response as above.)

Round 2
Reviewer 1 Report
The authors have adequately addressed my comments.